Gamebird responses to anthropogenic forest fragmentation and degradation in a southern Amazonian landscape

Michalski Fernanda fmichalski@unifap.br 1 2
Peres Carlos A. 3
1 Laboratório de Ecologia e Conservação de Vertebrados, Programa de Pós-Graduação em Biodiversidade Tropical, Universidade Federal do Amapá , Macapá , Amapá , Brazil
2 Instituto Pró-Carnívoros , Atibaia , São Paulo , Brazil
3 Centre for Ecology, Evolution and Conservation, School of Environmental Sciences, University of East Anglia , Norwich , Norfolk , United Kingdom
Pimm Stuart
Electronic publication date: 2017 Jun 7
Publication date: 2017
Volume: 5
Electronic Location ID: e3442
Received 2017 Mar 21; Accepted 2017 May 18
Copyright: ©2017 Michalski and Peres
Copyright year: 2017
Copyright holder: Michalski and Peres
License: This is an open access article distributed under the terms of the Creative Commons Attribution License, which permits unrestricted use, distribution, reproduction and adaptation in any medium and for any purpose provided that it is properly attributed. For attribution, the original author(s), title, publication source (PeerJ) and either DOI or URL of the article must be cited.
License URL: https://creativecommons.org/licenses/by/4.0/

Keywords: Bird, Disturbance, Hunting, Logging, Wildfires, Tropical forest

Funding: NERC (UK) 2001/834 Center for Applied Biodiversity Sciences of Conservation International John Ball Zoological Society provided supplementary funds Brazilian Ministry of Education (CAPES) CNPq Process 301562/2015-6 This study was funded by a NERC (UK) grant (2001/834) awarded to CAP. The Center for Applied Biodiversity Sciences of Conservation International and the John Ball Zoological Society provided supplementary funds. The data presented here was collected during FM’s doctoral study, which was funded by a studentship from the Brazilian Ministry of Education (CAPES). FM receives a productivity scholarship from CNPq (Process 301562/2015-6). The funders had no role in study design, data collection and analysis, decision to publish, or preparation of the manuscript.

==============================
Although large-bodied tropical forest birds are impacted by both habitat loss and fragmentation, their patterns of habitat occupancy will also depend on the degree of forest habitat disturbance, which may interact synergistically or additively with fragmentation effects. Here, we examine the effects of forest patch and landscape metrics, and levels of forest disturbance on the patterns of persistence of six gamebird taxa in the southern Brazilian Amazon. We use both interview data conducted with long-term residents and/or landowners from 129 remnant forest patches and 15 continuous forest sites and line-transect census data from a subset of 21 forest patches and two continuous forests. Forest patch area was the strongest predictor of species persistence, explaining as much as 46% of the overall variation in gamebird species richness. Logistic regression models showed that anthropogenic disturbance—including surface wildfires, selective logging and hunting pressure—had a variety of effects on species persistence. Most large-bodied gamebird species were sensitive to forest fragmentation, occupying primarily large, high-quality forest patches in higher abundances, and were typically absent from patches <100 ha. Our findings highlight the importance of large (>10,000 ha), relatively undisturbed forest patches to both maximize persistence and maintain baseline abundances of large neotropical forest birds.

Introduction

Habitat loss and fragmentation are often considered to be the most serious threats to biological biodiversity. However, faunal assemblages in fragmented tropical forest landscapes not only become stranded in small isolated habitat patches, but are also subject to other anthropogenic disturbances, such as hunting (Peres, 2001), selective logging (Michalski & Peres, 2005), and wildfires (Cochrane & Laurance, 2002). These disturbances operate hand-in-hand with habitat fragmentation and can compound fragmentation-induced ecological effects on faunal communities (Laurance & Useche, 2009).

Among all the structural features of landscapes that can change with habitat loss and fragmentation, habitat size and isolation are usually considered the most important predictors of species persistence (Fahrig, 2003). However, subsistence, market and recreational hunting can lead to population declines or local extinctions even in large, well-connected forest patches (Michalski & Peres, 2005; Michalski & Peres, 2007), simplifying the species composition and size structure of residual faunal assemblages (Peres, 2001; Urquiza-Haas, Peres & Dolman, 2011). Forest wildfires, for example, affect the species richness and abundance of understory birds (Barlow & Peres, 2004), which may fail to recover even after 10 years of forest regeneration (Mestre, Cochrane & Barlow, 2013). Similarly, logged forests are impoverished of many forest specialist species, following population declines or local extinctions (Bicknell & Peres, 2010; Michalski & Peres, 2005; Michalski & Peres, 2007).

Patterns of habitat patch occupancy of tropical forest vertebrates in fragmented landscapes are highly variable, with some species more likely to persist than others (Gascon et al., 1999; Laurance et al., 2011; Michalski & Peres, 2007; Michalski & Peres, 2005). Pinpointing which species are most prone to local extinction and understanding the patch and landscape features that govern species persistence are therefore key conservation issues that remain poorly understood.

Large terrestrial gamebird species are vulnerable to both habitat disturbance and hunting pressure (Peres, 2001; Thiollay, 1999; Thiollay, 2005; Urquiza-Haas, Peres & Dolman, 2009; Urquiza-Haas, Peres & Dolman, 2011), and may be extirpated or become rare in isolated forest patches (Robinson & Robinson, 1999). The Alagoas Curassow (Mitu mitu) (Linnaeus, 1766), which now survives only in captivity (BirdLife International, 2016), is a classic example of a large neotropical bird that was driven to global extinction in the wild by the combined effects of habitat fragmentation and hunting (Bianchi, 2006). Many other forest gamebird populations continue to be hunted within habitat remnants across all major tropical forest regions (Brooks et al., 2001), with unknown consequences to population persistence. Yet few studies have examined the environmental and anthropogenic drivers of large-bodied bird extinction across human- fragmented forest landscapes. Because of the site-specific nature of avian responses to fragmentation (Brooks, 2006; Sigel, Robinson & Sherry, 2010), more data are needed to understand how any given species copes with environmental changes in fragmented landscapes (Thornton, Branch & Sunquist, 2012).

The Brazilian Amazon encompasses the most extensive tropical forest region remaining within a single country, but has experienced the highest tropical deforestation rate since the 1970s, particularly in its most accessible eastern and southern portions (INPE, 2016; Michalski, Peres & Lake, 2008; Skole & Tucker, 1993). These subregions are characterized by highly fragmented landscapes, containing thousands of forest patches of varying size, shape, and degree of disturbance under multiple disturbance regimes (Peres & Michalski, 2006; Prist, Michalski & Metzger, 2012). Here, we examine the effects of forest fragmentation and forest degradation on the persistence of a group of large-bodied terrestrial and canopy birds in a highly fragmented forest landscape of the southern Brazilian Amazon using both interview and line-transect censuses data. First, we describe levels of species persistence within remaining forest patches using standard species–area relationships. We then evaluate how these habitat patch area effects can interact with levels of human perturbation including hunting pressure, and consider how individual species traits may affect patterns of patch occupancy. Finally, we validate our interview data based on data obtained from a standardized series of line-transect censuses and make some general points about how patch- and landscape-scale habitat metrics can explain patterns of local extinction in gamebirds in highly fragmented tropical regions, such as the ‘deforestation arch’ of the Amazon.

Figure 1 Location of the study region in Alta Floresta, northern Mato Grosso, Brazil.

Location of the study region in Alta Floresta, northern Mato Grosso, Brazil (inset map), showing the 129 surveyed forest patches (black areas) and 15 continuous forest sites defined as ‘pseudo-controls’ (solid circles). Land cover analysis was based on a classified Landsat image at the time of the study (year 2001). Green, white, and blue areas represent forest, non-forest area (mostly cattle pastures), and open water, respectively.

Methods

Study area

This study was conducted in the region of Alta Floresta, located in northern Mato Grosso, southern Brazilian Amazonia (09°53′S, 56°28′W; Fig. 1). Large-scale deforestation in this landscape resulted from an agricultural resettlement scheme dating from the late 1970s. As of 2004, only 42% of the original forest cover remained in the Alta Floresta region (Michalski, Peres & Lake, 2008), which further declined to 35% by 2016, reflecting the more recent decelerating rate of absolute deforestation in this region (Fig. S1). This resulted in a hyper-fragmented landscape containing forest patches of varying size, shape and levels of structural and nonstructural forest disturbance, surrounded by an open-habitat matrix dominated by managed cattle pastures (Peres & Michalski, 2006). All 129 forest patches (mean ± SD patch size = 510 ± 1,658 ha, range = 0.47–13,551 ha) included in the study were semi- or entirely isolated from the skeletal forest matrix surrounding the wider Alta Floresta region. These were compared with 15 neighboring continuous primary forest, which we define here as ‘pseudo-control’ sites (Fig. 1). As the Alta Floresta region is a highly deforested landscape, even our continuous ‘pseudo-controls’ sites do not exactly represent pristine primary forest, thereby serving as a conservative baseline of the overall observed patterns. All sites were located within a 50-km radius of the town of Alta Floresta (09°54′S, 55°54′W) and were accessible by river, paved or unpaved roads, or both. Locations of all forest patches sampled were plotted using GPS coordinates obtained in situ, and are available in Appendix S1–S2.

Study design

Forest patches were initially selected using a georeferenced 2001 Landsat ETM image (scene 227/67), on the basis of their size, degree of isolation, and nature of the surrounding habitat matrix. Forest remnants in private landholdings were either entirely isolated by open cattle pasture or thinly connected to other patches by riparian forest corridors along perennial streams as required by Brazilian law. As a key prerequisite, all candidate sampling sites that had been previously identified using the georeferenced image were associated with at least one local informant, usually a long-term resident or landowner, who was (i) willing to be interviewed, (ii) an assiduous visitor to that previously selected patch, and (iii) thoroughly familiar with both the history of human disturbance of the patch and the medium to large-bodied gamebirds persisting within that patch. For each forest patch, supplemental information on both extrinsic and intrinsic disturbance stressors and its gamebird species were obtained from additional informants if the patch was shared and/or visited by neighboring landowners. Interviewees had been living next to, and often working within, each forest patch for at least five years (mean ± SD = 12.3 ± 7.8 yrs; N = 144), and regularly entered the patch for a range of semi-subsistence, extractive and/or recreational reasons. In aggregate, our interview methodology recovered species occupancy and forest disturbance data referring to 129 forest patches and 15 “pseudo-controls”.

Gamebird surveys

Interview data—From June to September 2001 and May to July 2002, we obtained a total of 144 interviews that resulted in patch occupancy data on the midsized to large-bodied gamebirds of each patch and “pseudo-controls”, including three cracids (Mitu tuberosum (Spix, 1825), Penelope jacquacu Spix, 1825, and Pipile cujubi (Pelzeln, 1858)), two tinamids (Tinamus major (Gmelin, 1789), and Tinamus tao Temminck, 1815), one psophiid (Psophia viridis Spix, 1825) and a woodquail (Odontophorus gujanensis (Gmelin, 1789)) that were widely familiar to hunters and other forest users in this region. Congeners that could not always be clearly distinguished (i.e., the large-bodied tinamids, Tinamus major and T. tao) were pooled under a single functional group (Table 1).

Table 1 Encounter rates and occupancy data on gamebirds across the southern Amazonian fragmented forest landscape surrounding Alta Floresta, Mato Grosso, Brazil.

Forest patch area, sampling effort, and bird species encounter rates obtained from line-transect walks (ER—detections per 10 km walked), and local occupancy data obtained from interviews (OCC) within 21 forest patches and two continuous forest sites surveyed across the landscape around Alta Floresta, Mato Grosso, Brazil.

Site	Area (ha)	TDLTa	Mitu tuberosum	Penelope jacquacu	Pipile cujubi	Psophia viridis	Odontophorus gujanensis	Tinamus spp.b	
			ER	OCC	ER	OCC	ER	OCC	ER	OCC	ER	OCC	ER	OCC	
151	2.4	5.39	0	0	0	0	0	0	0	0	0	0	0	0	
156	4.1	6.36	0	0	0	0	0	0	0	0	0	0	0	0	
143	4.6	6.37	0	0	0	0	0	0	0	0	0	0	0	0	
153	4.7	7.55	0	0	0	0	0	0	0	0	0	0	0	0	
7	6.6	8.32	0	0	0	0	0	0	0	0	0	0	0	0	
154	7.3	7.76	0	0	0	0	0	0	0	0	0	0	0	0	
42	14.9	14.41	0	0	0	0	0	1	0	0	0	0	0	1	
145	16.0	20.18	0	0	0	0	0.50	0	0	0	0	0	0	0	
94	21.5	17.24	0	0	0	1	0	0	0	0	0	0	0	1	
131	24.5	24.05	0	1	7.07	1	1.25	1	0.42	1	0	1	0	1	
152	25.7	15.41	0	0	3.24	1	0	0	0	0	0	0	0	0	
26	86.9	59.45	0	0	2.86	1	0	1	2.52	1	0	1	1.35	1	
67	98.2	56.20	0.18	1	3.74	1	0.18	1	4.27	0	0.18	0	0	1	
61	106.2	57.28	0	0	2.62	1	0	0	0.87	0	0	0	0.17	0	
114	141.3	51.24	0	0	3.12	1	0	1	2.15	0	0	1	0	1	
48	211.7	85.49	0.12	1	3.04	1	0.23	0	3.63	1	0	1	0	1	
29	787.2	114.78	0.09	0	1.48	1	0	1	4.27	1	0.09	1	0.26	1	
62	899.8	127.38	0.39	1	1.88	1	0	1	2.51	0	0	1	0.24	1	
19	1,763.3	157.06	0.06	1	2.67	1	0.19	1	1.27	1	0.13	1	0.13	1	
27	11,034.7	247.45	0.24	1	2.22	1	0.24	1	2.47	1	0.36	1	0.36	1	
16	14,480.5	211.45	1.66	1	2.41	1	0.57	1	3.74	1	0.19	1	0.47	1	
150	144,805.0	228.60	0.66	1	2.23	1	0.44	1	1.97	1	0.96	1	0.52	1	
155	144,805.0	210.31	15	1	2.23	1	0.48	1	1.90	1	0.10	1	0.33	1	
Notes.

a Total Diurnal Line Transect (TDLT) sampling effort (km walked).

b Includes both Great Tinamou (Tinamus major) and Grey Tinamou (Tinamus tao).

Occupancy records specifically referred to occurrences within a clearly defined forest patch that was usually within sight from the interviewee’s household. In all cases, interviews were aided by color plates in field guides, photographs, and recordings or imitations of vocally conspicuously species. To assess the frequency of type II errors, interviewees were asked to identify which species were present in the patch from a selection of gamebird color plates, including five species known to be entirely absent from the study region. In all interviews, interviewees never falsely identified large-bodied bird species known to occur only in other neotropical forest regions as present in their forest patches, which gives us confidence that the occupancy data they provided for these species were reliable. Occupancy records of a given species were defined as patch-level occurrences when interviewees had no doubt as to whether the species was locally present at the time of interviews or recent past, whether the species was thought to be a full-time resident or an occasional transient within the patch. Local extinction events were conservatively defined as unambiguous absences when a species had been reported to have once occurred within a patch but had not been sighted, heard or detected for at least two years.

Line-transect surveys—Our interview-based sampling protocol was further validated using a standardized line-transect protocol (Peres & Cunha, 2012). Terrestrial and arboreal mid-sized to large-bodied gamebirds were surveyed in a smaller subset of 15 forest patches and a single continuous forest sites in June–December 2003 and June–December 2004 (Michalski & Peres, 2007). An additional subset of six forest patches and one continuous forest site, that had not been sampled using interviews in 2001–2002, were surveyed during the following two years using both line-transect census and camera-trapping. However, occupancy data on gamebirds based on local interviews were also obtained for these patches following the same protocol and standard methods used in our original series of 144 standardized interviews. At least three observers simultaneously walked at least three transects (mean velocity ∼1,250 m/h) during rainless weather in the morning (06:30–09:30h) or afternoon (14:30–17:30h). Transects were surveyed during nonconsecutive days at least nine times within a 30-day period (Michalski & Peres, 2007). Any confounding effects of seasonality were minimized by systematically rotating our monthly census schedule across sites in different size classes and levels of disturbance. In total, our sampling effort amounted to 1,739.6 km of census walks, including 841.8 km and 897.8 km in forest patches <10,000 ha and >10,000 ha, respectively (Table 1). To minimize possible detectability bias related to patch size, all outlier visual and acoustic detection events (e.g., defined as perpendicular distances (PD) ⩾ 50 m) were excluded from the analysis. Sampled forest sites were highly variable in size, which resulted in inevitable between-site differences in cumulative census effort. As a consequence, the total area effectively sampled in small patches per patch area was proportionally larger than that in large patches. Therefore, our results should be considered conservative with respect to area effects.

Landscape metrics and anthropogenic disturbance

A suite of landscape variables were extracted from the image using Fragstats v. 3.3 (McGarigal & Marks, 1995) and ArcView 3.2. Following a two-stage unsupervised classification of the Landsat image (ETM 227/067, date 27th September 2001), it was possible to unambiguously resolve 10 mutually exclusive land cover classes including closed-canopy forest, semi-open forest, open-canopy forest, disturbed forest, highly disturbed forest, managed and unmanaged pasture, recent clear-cuts, bare ground, and open water. The image was georeferenced using a guide file from the Brazilian Space Agency (INPE) with an accuracy of 0.29 pixel, each of which with a resolution of 15m. For forest patches surveyed that were not completely isolated (60 of 129 patches), we artificially eroded the narrowest connections (mean width ± SE = 73.5 ± 5.9 m)—usually consisting of riparian corridors to other forest patches—in order to calculate the total patch area. Erosion of connections was always carried out across the narrowest groups of pixels representing the most disturbed class of forest cover such as young second growth. For each forest patch, we measured the patch size; the straight-line distance to Alta Floresta, defined as the distance from the urban centre to the nearest edge of each patch; and the forest habitat quality, expressed as the percentage of closed-canopy and semi-open forest pixels contained in each patch. For ‘pseudo-control’ continuous forest sites, which were broadly connected with the surrounding forest matrix beyond the boundary of our Landsat scene, we assigned an arbitrary forest area value of one logarithmic magnitude greater than our largest fragment (Table 1). Some minor differences in forest patch area between the interview data (obtained in 2001–2002) and line-transect census data (obtained in 2003–2004) are expected as patch area for the latter was calculated using a 2004 Landsat image (Michalski & Peres, 2007). To enable realistic estimates of patch size, patches that were still tenuously connected to any surrounding forest were artificially eroded on the basis of the narrowest, most disturbed forest cover connection at the time line-transect surveys were carried out. As an independent measure of patch connectivity, we conducted a cost surface (CS) analysis using ArcView 3.2 that considers a diffusion coefficient described by the pixel-specific habitat resistance to forest vertebrate dispersal to the nearest source areas. We assumed that these were located in the large blocks of continuous forest to the north and south of Alta Floresta, and assigned the lowest CS weight (1) to areas of either closed-canopy or semi-open forest, followed by highly disturbed and open-canopy forest (3), and unmanaged regenerating wooded pastures (5). The highest CS weight (30) was assigned to non-forest pixels (water, managed pasture and bare ground). We then calculated the overall CS value based on a procedure similar to a Euclidean distance function. However, cost surfaces were determined by the shortest non-linear and often sinuous cost distance (or accumulated travel cost) from each cell to the nearest source cell, rather than the linear distance between a patch and any potential source. Cost surfaces thus took into account both the length of connecting pixels available for animal movement along a path of least resistance and the forest habitat quality of those pixels for forest vertebrates. Small CS values represent low dispersal costs from source areas to well-connected patches, usually through suitable riparian forest corridors, and high values represent high dispersal costs to poorly connected or unconnected patches, usually through non-forest areas. Forest isolate age was obtained from the interview data and cross-validated with a series of Landsat images (Michalski, Peres & Lake, 2008) and is defined as the number of years since the surrounding open habitat matrix had been formed by extensive clear-cutting of adjacent primary forest areas, thereby isolating the patch.

The intensity and extent of disturbance within each forest patch was ranked using a five-point scale (0–4) on the basis of site inspections and information obtained from interviewees according to (i) the spatial scale of selective logging activities, (ii) severity and proportional coverage of ground fires, which typically penetrated into forest patches from adjacent pastures, and (iii) degree of hunting pressure. Our measure of logging intensity took into account the method of timber felling and removal (and thus collateral damage to the residual stand), timber species selectivity, the amount and relative extent of timber extraction, and the number of years since logging practices had been discontinued. The scale of burn severity considered the proportion of each forest patch area that had been burned (interior and edge) prior to interviews, the intensity of the surface fire, and the number of recurrent fires. Information obtained on the level of subsistence and recreational hunting pressure within each patch included the frequency and number of years of game exploitation, the number of hunters using that patch, and whether hunting activities were still taking place at the time of interviews. Subsistence hunters throughout the Alta Floresta region were both uninhibited by interviewers and largely unaware of legislation concerning game hunting regulations designed to protect threatened species which, in any case, are rarely enforced in Brazilian Amazonia by the Environment and Renewable Natural Resources Agency (IBAMA). The complete lack of local enforcement of hunting restrictions therefore did not encourage potentially dishonest information on hunting practices, so we consider this potential source of bias negligible.

Data analysis

All analyses were performed in R (R Development Core Team, 2015). To examine the patch size characteristics in relation to the presence or absence of six gamebird species and test for differences in mean sizes of occupied and unoccupied patches we used two-sample t-tests. To assess the species-area relationships between forest patch area (log 10 ha) and number of gamebird species we performed linear regression models considering all 129 forest patches. The r2 value that we report is always the adjusted r2. To assess the patch occupancy probability for each species, we examined the effects of (i) patch variables (e.g., patch size, habitat quality, isolate age), (ii) landscape variables (e.g., CS value, distance to Alta Floresta), and (iii) levels of human disturbance within a patch (burn severity, logging intensity, and hunting pressure) as independent variables in logistic regression models. We controlled for high levels of inter-dependence between patch and landscape variables by performing a Pearson correlation matrix, and excluding those variables that were intercorrelated by |R| > 0.70. A number of additional patch metrics were also calculated during the analysis (e.g., edge-to-area ratio, core patch area >100 m from nearest edge) but these were strongly correlated with forest patch area (r > 0.80). We also calculated the proportion of closed-canopy forest area contained within circular buffers of 1 and 2 km from the geometric center of each forest patch, but these landscape variables were also highly correlated with log patch area (1-km buffers: r = 0.79; 2-km buffers: r = 0.66, N = 129, p < 0.001). We used generalized linear models (GLMs) with a binomial (logit) error structure to investigate predictors of species occurrence, with presence or absence of the six gamebirds in each forest patch as response variables. As predictors we used patch size, habitat quality, isolate age, CS value, distance to Alta Floresta, burn severity, logging intensity, and hunting pressure. The influence of these predictors on the response variables was tested with separate GLMs to understand how these predictors could affect the occurrence of each species. To improve numerical stability of the GLMs the continuous variables were standardized (centered and scaled by their standard deviation). To identify the best predictors of gamebird occurrence, we adopted an information theoretic model averaging framework based on Akaike’s information criterion (AIC) (Burnham & Anderson, 2002). We developed a set of six a priori candidate models to represent the effects of patch area, forest habitat quality and anthropogenic perturbations on species occupancy. Alternative candidate models in each set were compared using the difference in their AICc (corrected Akaike Information Criterion for small sample sizes) values in relation to the first-ranked model (ΔAICc) (Burnham & Anderson, 2002) and implemented in the ‘MuMIn’ R package (Barton, 2016). A value of ΔAICc ≤2 indicates equally plausible models (Burnham & Anderson, 2002). Relative likelihood of each model was estimated with Akaike weights (ωi) (Burnham & Anderson, 2002), which ranges between 0.0 (least important) and 1.0 (most important). This approach enables multi-model inference (MMI), which is considered to have numerous advantages over traditional hypothesis testing of a single null model (Burnham & Anderson, 2002; Grueber et al., 2011).

We analyzed the line-transect census data in terms of single animal (or group) and detection rates as the number of encounters per 10 km walked. When conducting line-transect census we accounted for differences in groups (flock) size, even though some species were either solitary or foraged in pairs. Therefore, our abundance estimates are based on reliable flock counts or mean site-specific flock size for all encounter events at any given site for which it was not possible to count or estimate flock size (Peres, 1999; Peres & Cunha, 2012). We obtained those occupancy and abundance data for the same six gamebird taxa considered during local interviews. We then plotted abundance (detection rate) against forest-patch area (log10x) in order to evaluate area-effects. Abundance data were log-transformed (log10x + 1) to improve normality.

Results

Species persistence and area effects

The size distribution of the 129 forest patches surveyed on the basis of interviews ranged from 0.47 to 13,551 ha (mean ± SD = 510 ± 1,658 ha) and these patches had been isolated for 4 to 27 years (mean ± SD = 15.3 ± 5.9 years). The subset of 21 forest patches surveyed using line-transect censuses ranged in size from 2.4 to 14,481 ha (mean ± SD = 1,416.3 ± 3,741.4 ha) and from a few months to 27 years of post-isolation at the time of surveys (mean ± SD = 16.0 ± 8.1 years). These patches retained between zero and all six gamebird species considered here, with a mean of 3.0 ± 2.0 species based on interviews, and 2.7 ± 2.3 species based on surveys on foot (Table 1). In contrast, the 15 continuous forest sites surveyed contained 4–6 species (5.5 ± 0.8 species) based on interviews, and all six species on the basis of line-transect censuses (Table 1).

Gamebirds varied widely in their rates of patch occupancy ranging from as many as 67% of all patches surveyed for Tinamus spp. (including Tinamus major or Tinamus tao) to as few as 30-32% for Psophia viridis and Mitu tuberosum. Occupied forest patches were significantly larger than unoccupied patches for all but one gamebird species (Table 2), as there was no significant patch area difference for Pipile cujubi (Table 2). On the basis of census data alone, all gamebird populations had been extirpated from all six forest patches smaller than 10 ha.

Figure 2 Relationship between forest patch area and gamebird species richness within 129 forest patches and 15 continuous forest sites.

Relationship between forest patch area (log10 ha) and gamebird species richness (A: All, B: Cracid species) within 129 forest patches and 15 continuous forest sites (solid circles). Colors represent the gradient of habitat quality (HQ). Linear regression lines (mean) and shaded areas (±95% CI) were obtained from model predictions.

Table 2 Occupied and unoccupied forest fragments for six Amazonian gamebird species.

Size of occupied and unoccupied forest patches for all six bird taxa examined in this study.

Species	t valuea	df	Forest patch size (ha)	
			Occupied patches	Unoccupied patches	
			N	Rangeb	Mean ± SD	N	Rangec	Mean ± SD	
Mitu tuberosum	−4.100***	127	41	30–13,551	1,337 ± 2,773	88	0.5–998	125 ± 186	
Penelope jacquacu	−2.184*	127	84	5–13,551	740 ± 2,017	45	0.5–998	81 ± 167	
Pipile cujubi	−1.890+	127	74	6–13,551	745 ± 2,120	55	0.5–3,536	193 ± 508	
Psophia viridis	−3.364**	123	39	30–13,551	1,236 ± 2,837	86	0.5–3,536	187 ± 425	
Odontophorus gujanensis	−2.351*	123	64	9–13,551	862 ± 2,263	61	0.5–3,536	167 ± 471	
Tinamus spp.d	−5.302***	127	86	5–13,551	714 ± 1,998	43	0.5–998	101 ± 187	
Notes.

a Two-sample t-tests.

+ Not significant.

* p < 0.05.

** p < 0.01.

*** p < 0.001.

b Range of patch area (ha) with occurrences.

c Range of patch area (ha) without occurrences.

d Includes Tinamus major and Tinamus tao, which could not always be distinguished in the field.

Despite these species differences, there was a discernible species-area relationship for gamebird species across the Alta Floresta landscape. Forest patch area alone explained 45.8% of the variation in total species richness (p < 0.001), and 38.1% of the variation in the number of cracid species (p < 0.001) occurring in the 129 forest patches (Fig. 2). Likewise, gamebirds showed a clear abundance-area relationship, indicating a threshold patch area value of around ∼100 ha, below which most of the species were either absent or occurred in very low abundances (Fig. 3). A total of 49 of all 58 (84.4%) local extinctions observed within fragments (excluding continuous forest sites) for those six avian taxa occurred in patches <100 ha. However, these abundance responses to patch area were not evident in Pipile cujubi, increased sharply in Penelope jacquacu and Tinamus spp., and were relatively gradual in the other three taxa in patches larger than 100 ha.

Figure 3 Relationship between forest patch area and gamebird species abundance within 21 forest patches and two continuous sites.

Relationship between forest patch area (log10 ha) and abundance (log10x + 1, where x = encounters per 10 km walked) of six gamebird species (A–F) within 21 forest patches and two continuous forest sites. Lines (mean) and shaded areas (±95% CI) show predictions obtained from locally weighted non-parametric polynomial regression (loess).

Table 3 Model selection results of occurrence of six gamebird species in the landscape around Alta Floresta, Mato Grosso, Brazil.

Model selection results explaining the occurrence of six gamebirds within 129 forest patches in the fragmented forest landscape around Alta Floresta Mato Grosso, Brazil. All candidate models are GLM models with a binomial (logit) error structure.

Species’ models	Component models	
	df	logLik	AICc	ΔAICc	ωi	
Mitu tuberosum						
PA+HQ+CS+IA	5	−45.41	101.30	0.00	0.99	
PA+DAF+HQ+CS+IA+BS+LI+HP	15	−38.36	110.97	9.67	0.01	
PA	2	−54.42	112.94	11.64	0.00	
PA+DAF+BS+LI+HP	12	−44.11	114.90	13.60	0.00	
HQ+CS+IA	4	−57.61	123.54	22.24	0.00	
DAF+BS+LI+HP	11	−57.62	139.49	38.19	0.00	
Penelope jacquacu						
PA+HQ+CS+IA	5	−54.42	119.33	0.00	0.95	
PA	2	−60.96	126.01	6.68	0.03	
PA+DAF+BS+LI+HP	12	−50.63	127.94	8.61	0.01	
PA+DAF+HQ+CS+IA+BS+LI+HP	15	−48.02	130.28	10.95	0.00	
HQ+CS+IA	4	−66.58	141.49	22.15	0.00	
DAF+BS+LI+HP	11	−67.05	158.36	39.03	0.00	
Pipile cujubi						
PA+HQ+CS+IA	5	−72.43	155.35	0.00	0.76	
HQ+CS+IA	4	−74.83	157.98	2.64	0.20	
PA	2	−78.74	161.57	6.23	0.03	
PA+DAF+HQ+CS+IA+BS+LI+HP	15	−68.92	172.09	16.75	0.00	
PA+DAF+BS+LI+HP	12	−72.73	172.14	16.79	0.00	
DAF+BS+LI+HP	11	−78.11	180.48	25.14	0.00	
Psophia viridis						
PA	2	−61.75	127.60	0.00	0.67	
PA+HQ+CS+IA	5	−59.55	129.59	2.00	0.25	
PA+DAF+BS+LI+HP	12	−52.63	132.04	4.44	0.07	
PA+DAF+HQ+CS+IA+BS+LI+HP	15	−51.10	136.60	9.00	0.01	
HQ+CS+IA	4	−66.51	141.34	13.75	0.00	
DAF+BS+LI+HP	11	−63.86	152.05	24.46	0.00	
Odontophorus gujanensis						
PA	2	−70.25	144.59	0.00	0.88	
PA+HQ+CS+IA	5	−69.02	148.54	3.95	0.12	
PA+DAF+BS+LI+HP	12	−67.85	162.48	17.89	0.00	
HQ+CS+IA	4	−78.16	164.66	20.07	0.00	
PA+DAF+HQ+CS+IA+BS+LI+HP	15	−66.99	168.38	23.79	0.00	
DAF+BS+LI+HP	11	−80.54	185.43	40.83	0.00	
Tinamus spp.a						
PA	2	−68.94	141.98	0.00	0.80	
PA+HQ+CS+IA	5	−67.21	144.91	2.92	0.19	
PA+DAF+BS+LI+HP	12	−62.02	150.72	8.73	0.01	
PA+DAF+HQ+CS+IA+BS+LI+HP	15	−59.45	153.15	11.17	0.00	
HQ+CS+IA	4	−74.29	156.91	14.92	0.00	
DAF+BS+LI+HP	11	−69.59	163.44	21.46	0.00	
Notes.

PA Patch area (log10 ha)

DAF Distance from the Alta Floresta urban center

HQ Forest habitat quality

CS Cost surface value

IA Forest isolate age

BS Burn severity

LI Logging intensity

HP Hunting pressure

df degrees of freedom

logLik log-Likelihood of the model

AICc AIC value

ΔAICc difference in AICc value compared to the first ranked model

ωi Akaike weight; coefficients for each variable of the model

a Includes both Great Tinamou (Tinamus major) and Grey Tinamou (Tinamus tao).

Models containing forest patch area (either in a single model or in addition to habitat quality predictors) were also ranked as first order when predicting patterns of occupancy for all six gamebird species separately (Table 3). Overall, all models ranked as first order provided a high Akaike weight, ranging from 0.67 to 0.99 as strength of evidence, reinforcing that patch area was the strongest predictor of gamebird occurrence (Table 3). Additionally, patch area was a positive significant predictor of species occurrence for all species when conditional model-averaged slope coefficients were considered (Table 4). We thus examined the occupancy probability of each species considering only patch area, the most important predictor of species richness. Given the shape of logistic regression curves modelling occupancy, including both interview and line-transect census data, only one wide-ranging species (P. cujubi) failed to exhibit a sharp patch-area threshold (Fig. 4). Models containing forest patch area alone were again the first ranked models when predicting patch occupancy for Psophia viridis, Odontophorus gujanensis, and Tinamus spp. (Table 3). Forest habitat quality (expressed as the percentage of closed-canopy and semi-open forest pixels contained in each patch), cost surface and isolate age were also retained in first order ranked models with the additive effect of patch area for the three cracids (Mitu tuberosum, Penelope jacquacu, and Pipile cujubi) and large tinamous (Tinamus spp.) (Table 3). The conditional-model averaged coefficients for the same three cracids showed that, among the predictors used in a priori models, habitat quality was the key variable on the set of models with highest Akaike weight (Table 4).

Table 4 Conditional model-averaged slope coefficients (±SE) of predictors of the occurrence of gamebirds in the landscape around Alta Floresta, Mato Grosso, Brazil.

Conditional model-averaged slope coefficients (with associated ±SE in parentheses) of predictors of the occurrence of six taxa of gamebirds within 129 forest patches in the landscape around Alta Floresta, Mato Grosso, Brazil.

	Mitu tuberosum	Penelope jacquacu	Pipile cujubi	Psophia viridis	Odontophorus	Tinamus spp.a	
Patch area	1.67 (0.41)***	1.57 (0.38)***	0.56 (0.26)*	1.25 (0.32)***	1.27 (0.28)***	1.13 (0.28)***	
Distance from the Alta Floresta urban center	0.61 (0.40)	0.70 (0.32)*	0.34 (0.28)	0.18 (0.28)	0.19 (0.24)	0.39 (0.26)	
Forest habitat quality	1.32 (0.38)***	0.81 (0.31)*	0.79 (0.27)**	0.53 (0.29)†	0.36 (0.27)	0.48 (0.27)†	
Cost surface value	0.27 (0.31)	−0.21 (0.27)	−0.13 (0.22)	−0.07 (0.25)	−0.03 (0.23)	0.22 (0.24)	
Forest isolate age	−0.41 (0.27)	−0.30 (0.23)	−0.15 (0.20)	0.03 (0.24)	−0.01 (0.21)	−0.08 (0.21)	
Burn severity (compared with never burned)							
Light burn	0.37 (0.71)	−0.11 (0.67)	0.58 (0.55)	−0.15 (0.62)	−0.64 (0.54)	0.79 (0.59)	
Moderate burn	−0.08 (1.33)	−2.06 (1.10)†	−0.91 (0.88)	−0.82 (1.04)	−0.50 (0.90)	0.76 (0.92)	
Severe burn	−0.85 (1.42)	−0.39 (0.73)	0.02 (0.70)	−0.54 (0.79)	−0.03 (0.62)	1.76 (0.74)*	
Logging intensity (compared with unlogged)							
Light logging	1.72 (0.93)†	1.10 (0.88)	−0.42 (0.65)	−0.44 (0.76)	0.43 (0.65)	0.04 (0.70)	
Moderate logging	0.05 (0.90)	−0.24 (0.72)	−0.41 (0.60)	0.86 (0.74)	0.60 (0.60)	0.24 (0.64)	
Heavy logging	0.75 (1.37)	−0.18 (0.88)	−0.39 (0.79)	0.33 (0.90)	−0.08 (0.77)	−1.28 (0.81)	
Hunting pressure (compared with unhunted)							
Light hunting	0.26 (1.43)	−1.31 (0.99)	−0.11 (0.84)	−3.99 (1.36)**	−0.79 (0.84)	−0.42 (0.89)	
Moderate hunting	0.91 (1.21)	−0.96 (0.90)	−0.47 (0.77)	−1.88 (0.83)*	−0.72 (0.75)	0.14 (0.82)	
Heavy hunting	1.31 (1.29)	−1.06 (1.10)	0.53 (0.88)	−1.46 (0.92)	−0.96 (0.86)	0.71 (0.96)	
Notes.

a Includes both Great Tinamou (Tinamus major) and Grey Tinamou (Tinamus tao).

Significance levels:

† <0.10.

* <0.05.

** <0.01.

*** <0.001.

Figure 4 Probability of occurrence of six gamebird species in forest patches in the study region in Alta Floresta, northern Mato Grosso, Brazil.

Mean (±95% CI) occurrence probability of six gamebird species (A–F) within 129 (blue line) forest patches as a function of forest patch size, predicted using logistic regression models using species occupancy data based on local interviews (see text). Red lines show the same logistic functions based on 21 fragments surveyed using line-transect walks. Presence and absence data based on either interviews (black circles) or line-transect censuses (red circles) are presented for each species. All data from continuous forests (15 sites surveyed using interviews and two using line-transect censuses) were excluded from these models.

Considering different forms of patch-scale anthropogenic forest disturbance, two species were significantly affected by either burn severity (Tinamus spp.) or logging (Mitu tuberosum) (Table 4). Our metric of hunting pressure within a patch had a significant negative effect on the occurrence probability of only one gamebird species (Psophia viridis) (Table 4).

Discussion

This study across a large fragmented forest landscape of southern Amazonia showed that (1) forest patch area is the strongest determinant of gamebird species persistence and abundance; (2) forest habitat quality also has a positive effect on gamebird species persistence; and (3) the impact of different patterns of anthropogenic disturbance is widely variable across species. The importance of habitat patch size and quality has already been well documented for other groups of mid-sized and large bodied vertebrates in tropical forests (Michalski & Peres, 2005; Michalski & Peres, 2007; Thornton, Branch & Sunquist, 2012; Urquiza-Haas, Peres & Dolman, 2009), but interactions with habitat disturbance have been largely overlooked by other studies.

There is no evidence to suggest that differences in species composition between sites were due to pre-existing differences in forest types and tree species composition (Michalski, Nishi & Peres, 2007), and all species of large birds considered here are forest habitat generalists that are widely distributed throughout the Alta Floresta region (Lees et al., 2008; Lees & Peres, 2006). Additionally, as recently as 1976, the Alta Floresta region was entirely covered by undisturbed Amazonian terra firme forest of similar physiognomy (Michalski, Peres & Lake, 2008; Oliveira de Filho & Metzger, 2006). Thus, we cannot attribute patch-level differences in species composition to different patterns of forest habitat specificity, and can assume that in the past most species occurred at all sites surveyed. We first turn to conventional area effects, and then explore how other variables interacted with patch area to determine levels of species persistence.

Effects of patch size

Our results show that forest patch area was the strongest predictor of species occurrence, explaining as much as 46% of the overall variation in species richness across all patches. Occupied patches were significantly larger than unoccupied patches for five of the six gamebirds considered here. This is confirmed by the absence of mid-sized to large-bodied gamebirds in all forest patches surveyed using line-transect censuses, with a clear threshold of around 100 ha, below which encounter rates were very low when compared to larger forest patches and continuous forest sites. Similar trends were also confirmed by a camera-trapping survey conducted at the same forest sites at the time of the line-transect surveys (F Michalski, 2004, unpublished data). The collapse of avifaunal assemblages across a wide body size spectrum has also been shown for small forest patches in the Alta Floresta landscape (Lees & Peres, 2008a). This is consistent with the vulnerability of several large-bodied birds to forest loss and fragmentation in other neotropical landscapes (Thornton, Branch & Sunquist, 2012). This may be due not only to their large size, which is often associated with fragmentation-induced susceptibility to extinction in tropical birds (Burney & Brumfield, 2009; Kattan, Alvarezlopez & Giraldo, 1994), but also the associated frugivore-granivore diet and forest dependency (Vetter et al., 2011), which likely renders these species more prone to food resource scarcity and patch-scale extinctions in human-modified forest landscapes.

The only species that appear to be fairly insensitive to forest patch area, when compared to other sympatric gamebirds, was Pipile cujubi, a highly frugivorous and widely mobile canopy cracid (Galetti et al., 1997) that often traverses the open-habitat matrix of cattle pastures in our study region (Lees & Peres, 2009). This species can tolerate limited human disturbance and hunting pressure in landscapes dominated by small-scale agriculture as long as large fruiting canopy trees are left intact in nearby forest (Hayes et al., 2009). Due to the high vagility of this species across our highly fragmented study landscape, with large forest patches surrounding small and mid-sized patches, P. cujubi likely exhibited transient occupation of small forest patches as they trap-lined large fruiting trees, and much more so than the other species considered here. Moreover, large avian canopy frugivores are generally more adept at gap crossing than small understory insectivores (Lees & Peres, 2009; Thiollay, 1999), which ensures that even patches isolated by relatively large pasture areas continue to be used. Thus, the incidence of P. cujubi in forest patches <100 ha was likely represented by transient individuals that were capable of traversing wide gaps of pasture. This is consistent with avian surveys based on point-counts at 31 forest patches in the Alta Floresta landscape showing that this species was only detected at patches larger than 10,000 ha (Lees & Peres, 2006). Moreover, the occurrence probability of this species was lower based on only ∼30 days of census walks in each patch, compared with our interview data, which essentially averages occupancy records over much longer time frames. Indeed, this was the only species exhibiting marked differences in occurrence probability between the two survey methods we used. This highlights the prevalence of Type I errors (false absences) in relatively short field surveys, in that they can often fail to record transient individuals, in the absence of a stable resident population. In fact, the failure to record this often locally rare species within its known geographic distribution is not uncommon (Haugaasen & Peres, 2008).

Our logistic regression models showed that forest patch area was the strongest and most common predictor of species persistence, significantly affecting the presence of all six large-bodied birds. All species considered here could be described as either frugivores or granivores, but three species (P. jacquacu, Odontophorus gujanensis, and Psophia viridis) consume significant amounts of arthropods (Haugaasen & Peres, 2008). All species were therefore more prone to local extinctions in small forest patches regardless of main dietary mode, although P. jacquacu was found in patches as small as 15 ha (Lees & Peres, 2006). All six bird taxa studied here include fruit in their diets as a primary or secondary food resource, while Tinamus spp., P. jacquacu, and O. gujanensis also consume considerable amounts of invertebrates (Peres & Palacios, 2007). However, there was no clear pattern of dietary mode affecting species persistence or abundance in our fragmented landscape. Instead, occupancy pattern of gamebirds were sensitive to habitat area and habitat quality with some species using secondary/degraded forest whereas others were restricted to core primary forest (Lees & Peres, 2006; Lees & Peres, 2008b).

Effects of disturbance

Our results indicate that the recent human perturbation regime in each survey site was also a key determinant of patch occupancy rates. Forest habitat quality and levels of patch disturbance were all important determinants for the persistence of at least some gamebird species. The spectral quality of remaining forest habitat was an important determinant of patch occupancy for three species, particularly those that rely on high forest basal areas and a closed canopy, such as Mitu tuberosum, Penelope jacquacu, and Pipile cujubi (Lees & Peres, 2010; Thiollay, 1999; Thiollay, 2005; Thornton, Branch & Sunquist, 2012).

In terms of different forms of anthropogenic disturbance, the severity of surface wildfires significantly affected the distribution of tinamous. Logging practices in Alta Floresta are relatively unselective, targeting a wide range of commercially valuable timber species (CA Peres, pers. comm., 2001). However, logging intensity was only a marginally, rather than significant predictor, of the occurrence of M. tuberosum. In Central Guyana, the density of curassows was reduced in logged forest, compared with unlogged forests (Bicknell & Peres, 2010). Hunting pressure was a significant negative predictor of the probability occurrence of P. viridis. Birds become more important hunting targets where large mammals are depleted (Thiollay, 2005), which is a common phenomenon in fragmented landscapes such as the Alta Floresta region (Michalski & Peres, 2007). Hunter preference for large-bodied terrestrial bird species is widespread in the northern neotropics (Peres, 2001; Thiollay, 2005; Urquiza-Haas, Peres & Dolman, 2009), and unlike other Amazonian game vertebrates, such as primates, the colonist farmers of European descent who settled in northern Mato Grosso in the late 1970s to early 1980s have no hunting taboos against large terrestrial birds. Trumpeters are highly conspicuous group-living terrestrial frugivore-faunivores due to their alarm calls, rendering them easily detectable to hunters (Thiollay, 2005). They are also habitual followers of army-ant swarms, which tend to be reduced in density of driven to local extinction in small patches, which in turn are more likely to be overhunted, thereby suggesting that area-effects likely interacts with hunting pressure to drive this species to local extirpation in small patches.

Conservation implications

The large-bodied neotropical forest birds considered in this study were highly vulnerable to habitat loss and fragmentation. The deforestation history in the Alta Floresta region is relatively recent (∼40 years), so that the extensive patch occupancy data we present can substantially underestimate further population declines and local extinctions which will likely continue to take place in the near future. Moreover, the synergistic effects of reduced habitat area, a hostile intervening matrix, and elevated hunting pressure can result in additional extinction events of trumpeters and other large-bodied bird species (Peres, 2001; Thiollay, 2005). Despite the at least temporary persistence of some species in a modest number of small forest patches (<100 ha), our results highlight the conservation importance of retaining large tracts of relatively undisturbed primary forest to the original large vertebrate fauna in this region, including gamebirds. In addition, our local abundance data clearly showed a patch size threshold of around ∼100 ha, below which most gamebirds considered here failed to persist even at small population sizes. Therefore, even wide-ranging species exhibiting wide movements across the pasture matrix, such as Pipile cujubi, may not be able to cope with highly fragmented landscapes, if remaining patches continue to be hunted indiscriminately. Moreover, future extrapolations of population persistence from this study may err on the side of optimism as the forest habitat quality of existing forest patches is likely to be further degraded over time (Zimbres, Peres & Machado, 2017). Many large canopy tree species—that are relicts from the original old-growth flora thus providing important fruit sources to large-bodied birds—will likely succumb to high mortality and non-random replacements under edge-hyperdisturbed conditions (Tabarelli, Peres & Melo, 2012), further eroding the resource base of large frugivorous vertebrates. Additionally, most gamebird species assessed here can be effective seed dispersers in this fragmented landscape so that local extinctions can reduce sapling recruitment and alter relative abundances of animal-dispersed plants (Terborgh et al., 2008).

Conservationists will increasingly face daunting challenges in identifying which species are most extinction-prone in highly fragmented tropical forest landscapes, and which patch and landscape features will effectively minimize or retard extinction events. These are key conservation management issues that will need to be addressed if we are to prevent a widespread defaunation process across the patchwork of forest remnants persisting in these human-dominated landscapes.

Supplemental Information

Appendix S1 Interview raw data

Coordinates, forest patch area, distance from the Alta Floresta urban center, forest habitat quality, cost surface value, forest isolate age, burn severity, logging intensity and hunting pressure of six gamebird taxa within 129 forest patches and 15 ‘pseudo-controls’ surveyed with local interviews across the landscape around Alta Floresta, Mato Grosso, Brazil.

Click here for additional data file.

Appendix S2 Line transect censuses raw data

Coordinates, forest patch area, and records of field observations of six gamebird taxa within 21 forest patches and two ‘pseudo-controls’ surveyed across the landscape around Alta Floresta, Mato Grosso, Brazil.

Click here for additional data file.

Figure S1 Land-use maps representing the evolution of the landscape structure over a 12-year period in the Alta Floresta region of southern Brazilian Amazonia from (A) 2004 to (B) 2016

Land-cover classes are represented by forest (green), non-forest (white) and water (blue). Total forest cover in this study region declined from 42% in 2004 to 35% in 2016. This 7,295 km2 study landscape is bounded by the Teles Pires river to the north and east. Forest cover was not substantially reduced over this period, but the level of forest degradation apparently increased substantially.

Click here for additional data file.

Geraldo Araújo and Iain Lake provided invaluable assistance during the fieldwork and GIS analysis, respectively. We are grateful to Stuart Pimm, Daniel Brooks, Mariana Vale and one anonymous referee for comments on the manuscript.

Additional Information and Declarations

Competing Interests

Author Contributions

Data Availability

The authors declare there are no competing interests.

Fernanda Michalski and Carlos A. Peres conceived and designed the experiments, performed the experiments, analyzed the data, contributed reagents/materials/analysis tools, wrote the paper, prepared figures and/or tables, reviewed drafts of the paper.

The following information was supplied regarding data availability:

The raw data has been supplied as a Supplementary File.

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
