# Peer review of "Gamebird responses to anthropogenic forest fragmentation and degradation in a southern Amazonian landscape"

_PeerJ, doi:10.7717/peerj.3442_

## Round 0.1 · original submission · Major Revisions

It's particularly important to address all the comments, one by one, in the cover letter that you sent with the revision. None of these comments is a fatal criticism of your work, but there are a lot of them.

·

Basic reporting

Please use this reference in paragraph dealing with Alagoas Curassow:
Bianchi, C.A. 2006. Alagoas Curassow (Mitu mitu). Pp. 26-28 In: Conserving Cracids: the most Threatened Family of Birds in the Americas (D.M. Brooks, Ed.). Misc. Publ. Houston Mus. Nat. Sci., No. 6, Houston, TX.

as well as other studies that investigated similar Amazonian gamebird guilds, e.g.,:
Brooks, D.M., L. Pando-V., A. Ocmin-P., and J. Tejada-R. 2001. Resource separation in a Napo-Amazonian gamebird community. Pp. 213-225 In: Biology and Conservation of Cracids in the New Millenium (D.M. Brooks and F. Gonzalez-F., Eds.). Misc. Publ. Houston Mus. Nat. Sci. No. 2, Houston, TX.

and patterns of deforestation threatening Cracids, e.g.:
Brooks, D.M. 2006. The utility of hotspot identification for forest management: Cracids as bioindicators. Acta Zool.Sinica 52: 199-201.

Experimental design

I may have just missed it, but does it say anywhere in ms about # of visual vs. auditory detections? and were these weighted for analyses?

Again, I may have missed it, but does your ms mention about whether transect contacts were dealt with (weighted, etc) for observations of singletons vs multiples/flock? Psophia for example are invariably found in a flock, whereas most of the other species only for parent-offspring groups. In the latter situation, the association is much longer (9-12 months) is larger cracids than the smaller gamebirds. Was this dealt with in the analyses?

Validity of the findings

Highly important! Very nice work...

Additional comments

Really great work. I made some recommendations above re: citations and 'captures' along transects.

Just one general comment - Is this a typo in Fig 3 caption?
"loess regression"

Again, I want to commend you for this great work. Impressed as always!
Dan (Brooks)

·

Basic reporting

Fine. No comments.

Experimental design

Fine. No comments.

Validity of the findings

Fine. No comments.

Additional comments

The study investigates whether different stressors affect game bird occupancy and abundance in a tropical forest. The manuscript is very well written. I commend the authors for the quality of the field data and analysis, and for bringing new information from data that was collect long ago, and have already generated a number of relevant publications. The study includes a large number of likely predictors (patch metrics, habitat quality, fire, logging, and hunting) that have previously been evaluated for other organisms, but in isolation or in pairs. The main conclusion, however – that fragment size and disturbance level affects species’ occurrence – is not new (as the authors themselves recognize). Still, there is a serious lack of species-specific studies in the Tropics, as compared to temperate regions, that can provide quantitative data to support other, more comprehensive studies aiming at synthesis. The argument for carrying the study, however, can be improved.

Major comments:

An especially interesting result is the suggested 100 ha patch area threshold value, below which species are absent or occur at very low abundances. The authors insist that the pattern is clear, but the results, particularly Fig. 3, don’t seem to show a clear threshold at that area value (see details below).

Because the manuscript does not provide new insights, there is a need to better highlight the importance of studying this particular group of species. The “Conservation Implications” section is not particularly insightful. It states that the game birds are vulnerable to fragmentation and stress the beaten to death importance of retaining large tracts of relatively undisturbed forests. The six game species have relatively large ranges over the Amazon, and most are not threatened (although a few are Vulnerable under IUCN). Most species, however, are important seed dispersers. There are a number of studies showing that defaunation interferes with recruitment of late-succession, large seed trees.
Also, the study was carried in 2001, when Alta Floresta had > 40% forest cover (according to references provided by the authors). This forest cover was likely reduced in the last 12 years. The authors carried an unsupervised classification of a 2001 Landsat image. Why not doing the same for a current image to check how much forest cover the landscape has now? This would take very little time, and could provide interesting insights. Has Alta Floresta reached the forest cover threshold for bird community dismantling (30%, 10%, depending on the study)? Is it time for switching from conserving to restoring as a meaningful strategy in the arc of deforestation? Well, good seed dispersers are a key component of successful restoration projects. Many studies indicate that the Amazon is steadily approaching the 40% deforestation boundary at which it could switch to a new, savanna, equilibrium state. Anyhow, this is a neat study, with very solid data and analysis, but that could benefit from a more creative discussion of its implications.

Minor comments:

Line 99: The authors say that “As of 2004, only 42% of the original forest cover remained in the Alta Floresta region (Michalski et al., 2008).” Is there a more recent estimate of the percent forest cover in the region?

Line 86-88: Authors say: “First, we describe levels of species persistence within remaining forest patches using standard species-area relationships. We then evaluate how these effects can interact with levels of human perturbation (…).” It is not clear what “these effects” is referring to. Forest patch? Maybe its size? Or is it about species persistence? In that case it would not be an “effect”, thought. Please revise the wording to improve clarity.

Line 89: Authors say: “Finally, we validate our interview data based on data obtained from a standardized series of line-transect censuses.” There was no early mention to interview data. I suggest mentioning this method before informing how it was validated.

Line 106-107: Authors say “All sites were located within a 50-km radius of the town of Alta Floresta (GE: ‒9.876497, ‒56.086425) and were accessible (…).” Please explain what “GE” means. These seem to be just geographic coordinates in decimal degrees format. If that is the case, it would be more straightforward to report it in the conventional fashion, just as in Line 97. It would also keep consistency in the format used for coordinates throughout the text.

Line 181: For a better flow of ideas, I suggest presenting the patch metrics first (currently in Lines 197-202) and then the patch connectivity metrics (currently in Line 181-197).

Line 253-254: Revise the sentence. It seems to be missing words, and uses “in terms of” twice.

Line 260-265: The authors say that “The size distribution of the 129 forest patches surveyed on the basis of interviews ranged from 0.47 to 13,551 ha” and “The subset of 21 forest patches surveyed using line-transect censuses ranged in size from 2.4 to 14,481 ha”. If the 21 patches are indeed a subset, than the larger patch in that subset, with 14,481 ha, should be included in the 129 patches surveyed. Therefore, the larger patch among the 129 forest patches surveyed can’t be smaller than 14,481 ha (it was reported as having 13,551). Please check, or else provide more information that can explain this apparent incongruence.

Line 282: Correct “at in very low abundances”

Line 287-289: The authors say “only one species failed to exhibit a sharp patch-area threshold (Fig. 4).” Why not help out the reader here, by specifying in parenthesis which exceptional species is this (P. cujubi)?

Line 339: I suggest sticking with scientific names, instead of English names. PeerJ is not an ornithological journal, all graphs and most of the use scientific name.

Lines 350-353: The authors say that “the incidence of Piping Guans in forest patches <100 ha was likely represented by transient individuals that were capable of traversing the pasture matrix. This is consistent with avian surveys based on point-counts at 31 forest patches in the Alta Floresta landscape showing that this species was only detected at patches larger than 10,000 ha (Lees & Peres, 2006).” In fact, these two sentences seem at odd with each other (as opposed to consistent). In the same region, the current study finds the species in very small fragments, and in the other the species in only found in very large ones. The authors say that “Indeed, this was the only species exhibiting marked differences in occurrence probability between the two survey methods we used. This highlights the prevalence point Type I errors (false absences) in relatively short field surveys.” Still, this requires a bit more explanation. Lees & Peres (2006) study can be considered “short field surveys”? Is there an artifact arising from different sampling methods (transect vs. point-count)? Maybe from patch size distribution between the two studies? The current study was carried in 2001 and the other in 2004. Is that possible that the species disappeared from the small fragments in that period? Probably not. In any case, there is a need to better explain where this type I error comes from.

Lines 267-269: Authors say that “All species were therefore more prone to local extinctions in small forest patches regardless of main dietary mode, although P. jacquacu were found in patches as small as 15 ha (Lees & Peres, 2006).” I suggest discussing this result in light of Lees & Peres (2006). They found that “small-bodied flock-following primary-forest-dependent terrestrial insectivores” are highly sensitive to fragmentation. Can we suggest some general pattern for gamebirds? For big birds diet doesn’t matter? Or is just that they are mostly frugivorous, and frugivorours birds are less sensitive (independent of body size)?

Table 1: I infer that sites no. 150 and 155 are the two control sites (continuous forest) with line-transect survey. In the text (Line 200-202) the authors say “For ‘pseudo-control’ sites of continuous forest, which were broadly connected with the surrounding forest matrix beyond the boundary of our Landsat scene, we assigned an arbitrary forest area value of 100,000 ha.” In Table 1, however, both sites show an area of exactly 144,805 ha. It would be more accurate to say > 100,000 ha instaed, and indicate in the footnote that these are the continuous forest sites. Also, in the legend, where the authors say “within 23 forest sites surveyed”, I suggest saying “within 21 forest patches and two continuous sites”, as in Fig. 3.

Table 2: Please provide units for patches either in the heading or the legend (hectares, I supposed). The authors use “forest patch” more often throughout the manuscript, therefore, I suggest using that term in the legend (as opposed to “forest fragment”), for consistency. Or else, use “forest fragment” throughout the text.

Figure 1: The figure is nice and has been published before with the same basic design, but showing different patches. Still, I believe it could be improve by using green as opposed to grey for the continuous forest area, at least in the digital version of the paper (it would still be grey in the printed version). This way the authors would be able to say “black areas” in the legend, which is more understandable than “solid areas”. Also, because this one study has a lot of forest fragments, including some tiny ones, the black dots used to show the control sites are quite hard to spot. The use of a different color or symbols for the control sites would improve readability. A black border around the figure would also help. Finally, I suggest specifying in the legend the imagery source (Landsat) and year (2001, I suppose).

Figure 3: In Line 280-283 the authors say “Likewise, gamebirds showed a clear abundance-area relationship, indicating a threshold patch area value of ~100 ha, below which most of the species were either absent or occurred in at very low abundances (Fig. 3).” Figure 3, however, doesn’t show a clear, unique, trend. We see two obvious behaviors: (1) Tinamus sp. and Penelope jacquacu show a sharp increase in abundance up to 100 ha, and stabilization afterwards at about 3 individuals per 10 Km walked (if I interpreted the y-axis correctly, i.e [(100.5)+1]), and (2) the other four species, showing a much more smooth increase in abundance with patch area. Therefore, I believe that the interpretation of Fig. 3 provided in Lines 280-283 should be further detailed. Also, the axes labeling in Fig. 3 is a bit hard to follow. First, both axes use “x”, but “x” is a different variable in each axis (x = encounters/10 km walked in y-axis, and x = patch area in the x-axis). I suggest labeling x-axis as “Log10(Patch Area) (ha) (do the same for Fig. 2), and the y-axis as “Log10(Species Abundance) + 1”, and slightly re-phrasing the text legend, which already says that gamebird species abundance = encounters per 10 km walked. Also, it seems that in the graph there are some point data hidden underneath others. For Penelope jacquacu to have a sharp increase in abundance from <10 ha to 100 ha, for example, it must have a zero abundance point data somewhere <10ha. However, the lowest abundance point date we see in the graph for that species is at about 3 individuals (i.e [(100.6)+1]). I would guess that there is a P. jacquacu point data hidden underneath an O. gujanensis data points. Maybe the authors can use different symbols such as triangles for P. jacquacu, so that the reader can see different species that have the same data point value in the graph (if that is indeed the case). Also, correct “loess regression” in the legend.

Mariana Vale

Reviewer 3 ·

Basic reporting

1. BASIC REPORTING

This is a well-written, well-organized, clear and concise paper on the effect of anthropogenic disturbance on Amazonian gamebirds. The figures are slick and the conclusions and discussion are well justified with the data. My biggest issue is that of controlling for detectability, which in detail in the "validity of findings" section. Most of the remaining comments are quite minor.


ABSTRACT
24 What exactly are "local interview data". Interviews with hunters? A bit more detail would help the naive reader.

INTRO

86 Nicely described objectives. very clear

METHODS
fig 1 Not clear why you reference the river at the end of the fig legend

105 Why are these pseudo-controls? Please briefly explain this term.

143 Good thinking on this control

200 If these are forest birds, then why include semi-open forest as a measure of habitat quality? Generally I think calling this "habitat quality" is a little misleading, or at least a little vague. Why not just call it "proportion forest" or something along those lines?

249 backward stepwise selection may not be the best approach here. I prefer an a priori candidate set and model selection unless its an exploratory analysis.

RESULTS

261 Can you speak to the accuracy of the interview data for forest isolation age? I have seen other similar data from memory of interviewees that does not match landsat imagery very well at all. I can't recall if time since isolation was one of the explanatory variables that you explored in the analysis explicitly, but if it was, the accuracy of the data should be well justified (or caveats should be given).

266 interviews match surveys surprisingly well!

270 Seems to me that you should introduce these common names earlier on when you first mention the scientific names

Table 1: change first column 1 from "sites" to "site"

289-294 This should be in the methods section

300 Very surprising to me that hunting only affected this one species . . . I think this should be elaborated on in the discussion.

DISCUSSION

303 Nice summary of results - I prefer this approach to starting discussion

305 Perhaps parenthetically remind the reader how you defined habitat quality (most readers will skip or glaze through the methods). Ignore this if you choose to change your reference to "habitat quality" to "proportion forest" (or something similar), which I suggest you do.

347 Burney & Brumfield 2009 American Naturalist is a nice reference here for this pattern on longer timescales

358 Which is why it is critical to correct for detectability.

394 Cool observation . . .but source for this? or pers obs?

Experimental design

Well replicated; well designed. I'm convinced that the transects match the interviews quite well.

Validity of the findings

My biggest concern is related to the lack of control for detectability. Over the last ten years or so, the Patuxent group and many others (e.g. D. Mackenzie) have shifted the field towards analyses of occupancy and abundance that explicitly control for detectability through hierarchical modeling. Mackenzie's occupancy book and the R package PRESENCE are great resources that now have lots of online help and tutorials. Thus those logistic models can, with relative ease, be converted into detectability-controlled hierarchical logistic models. "occupancy" in the current lingo, refers to detectability-corrected occupancy rather than "naive occupancy" (not corrected for detectability), which is what is reported here as far as I can tell.

Along those same lines, the Patuxent folks as well as Mackenzie and many others have stressed that stepwise approaches such as the ones taken here are "data dredging", which risks spurious results. Alternatively, the Burnham & Anderson-type model selection approach, using AIC or a similar metric, in which competing a priori models are carefully compiled to reflect biological plausibility, tend to be preferred.

---

## Round 0.2 · accepted · Accept

I've read the reviews and your responses to them. Your paper is on a subject matter I know well for as you know I've published papers on the loss of species from Amazon habitat fragments. I think you've done a good job in your detailed response to the reviewers and so we should publish this revision. Thank you for considering PeerJ!